

# Faces in commonly experienced configurations enter awareness faster due to their curvature relative to fixation

Pieter Moors, Johan Wagemans and Lee de-Wit

Department of Brain & Cognition, Katholieke Universiteit Leuven, Leuven, Belgium

## ABSTRACT

The extent to which perceptually suppressed face stimuli are still processed has been extensively studied using the continuous flash suppression paradigm (CFS). Studies that rely on breaking CFS (b-CFS), in which the time it takes for an initially suppressed stimulus to become detectable is measured, have provided evidence for relatively complex processing of invisible face stimuli. In contrast, adaptation and neuroimaging studies have shown that perceptually suppressed faces are only processed for a limited set of features, such as its general shape. In this study, we asked whether perceptually suppressed face stimuli presented in their commonly experienced configuration would break suppression faster than when presented in an uncommonly experienced configuration. This study was motivated by a recent neuroimaging study showing that commonly experienced face configurations are more strongly represented in the fusiform face area. Our findings revealed that faces presented in commonly experienced configurations indeed broke suppression faster, yet this effect did not interact with face inversion suggesting that, in a b-CFS context, perceptually suppressed faces are potentially not processed by specialized (high-level) face processing mechanisms. Rather, our pattern of results is consistent with an interpretation based on the processing of more basic visual properties such as convexity.

## INTRODUCTION

The extent to which invisible stimuli are still processed has become a popular line of research over the last decades (*Dehaene & Changeux, 2011*; *Hesselmann & Moors, 2015*). One particularly compelling paradigm to render visual stimuli invisible is continuous flash suppression (CFS) (*Tsuchiya & Koch, 2005*). In CFS, a salient dynamic pattern composed of various colored shapes is presented to one eye while another stimulus is presented to the other eye. Due to the dynamic nature of the mask, the other stimulus is perceptually suppressed and invisible to observers for a time period on the order of seconds. CFS has been implemented in various ways to study processing of perceptually suppressed stimuli, one being the breaking CFS paradigm (b-CFS) (*Stein, Hebart & Sterzer, 2011*; *Gayet, Van Der Stigchel & Paffen, 2014*). Here, the contrast of the initially suppressed stimulus is gradually increased until it causes a perceptual breakthrough

Corresponding author
Pieter Moors,
pieter.moors@ppw.kuleuven.be

(i.e., becomes detectable to the observer). The breakthrough or suppression time is then used as an index of the strength of the representation of that visual stimulus during suppression. That is, as in regular binocular rivalry, 'stimulus strength' is predicted to influence suppression durations such that stronger stimulus representations break CFS faster than weaker stimuli (*Jiang, Costello & He, 2007*; *Stein, Hebart & Sterzer, 2011*). Here, it should be noted however that 'stimulus strength' is not a well-defined construct and that there is some debate as to which factors contribute exactly to differences in suppression times. That is, the factors driving breakthroughs could be low-level or high-level (*Gayet, Van Der Stigchel & Paffen, 2014*; *Hesselmann & Moors, 2015*), or involve the feature overlap between the CFS mask and the suppressed stimulus (*Yang & Blake, 2012*; *Moors, Wagemans & de-Wit, 2014*).

A number of studies have considered the degree to which face stimuli are still processed while perceptually suppressed and have used the b-CFS paradigm, amongst others, to tackle this question (for a review of unconscious face processing, not limited to CFS studies only, see *Axelrod, Bar & Rees, 2015*). A now-classic study by *Jiang, Costello and He (2007)* showed that upright face stimuli broke suppression faster than inverted face stimuli, resembling the well-known face inversion effect for consciously presented stimuli (*Yin, 1969*; *Farah, Tanaka & Drain, 1995*). Following this study, several b-CFS studies have replicated this face inversion effect (*Zhou et al., 2010*; *Stein, Hebart & Sterzer, 2011*; *Stein, Peelen & Sterzer, 2011*; *Stein & Sterzer, 2012*; *Stein, Sterzer & Peelen, 2012*; *Gobbini et al., 2013a*; *Gobbini et al., 2013b*; *Heyman & Moors, 2014*; *Stein, End & Sterzer, 2014*). Other studies have furthermore indicated that stimulus-related factors such as eye gaze (*Stein et al., 2011*; *Xu, Zhang & Geng, 2011*; *Chen & Yeh, 2012*; *Gobbini et al., 2013b*), facial expression (*Yang, Zald & Blake, 2007*; *Sterzer et al., 2011*; *Stein & Sterzer, 2012*; *Capitão et al., 2014*), face identity (*Geng et al., 2012*; *Gobbini et al., 2013a*), face race (*Stein, End & Sterzer, 2014*), or the trustworthiness or dominance of a face (*Stewart et al., 2012*) can influence suppression times. Taken together, these findings seem to suggest that, while perceptually suppressed, the representation of a face stimulus is a fairly integrated one involving the high-level analysis of several complex features.

In apparent contrast with these b-CFS findings, a more complicated pattern of results has arisen from studies that rely on adaptation to invisible face stimuli or investigate the representation of invisible face stimuli using neuroimaging techniques. For example, adaptation studies have indicated that visual awareness of a face is required for adaptation to complex features such as facial expression (*Yang, Hong & Blake, 2010*), face race or gender (*Amihai, Deouell & Bentin, 2011*), face identity (*Moradi, Koch & Shimojo, 2005*), face shape (*Stein & Sterzer, 2011*), or eye gaze (*Stein, Peelen & Sterzer, 2012*). The main conclusion of these studies is that adaptation effects for invisible stimuli are sometimes observed, but they are largely specific to the adapted eye and size of the stimulus. For example, *Stein and Sterzer (2011)* observed face shape aftereffects for fully invisible stimuli, yet these aftereffects were only observed if the test stimulus had the same size as the adaptor and was also presented to the same eye as the adaptor. This suggests that the adaptation occurred at a low level of processing, and was specific to simple features such as its exact size and shape. Similarly, neuroimaging studies have shown that neural responses

to invisible face stimuli are strongly reduced in the fusiform face area (*Jiang & He, 2006*; *Sterzer et al., 2014*), although the pattern of activation still enables the successful decoding of certain stimulus distinctions (*Sterzer, Haynes & Rees, 2008*; *Sterzer, Jalkanen & Rees, 2009*).

Taken together, behavioral studies relying on adaptation and neuroimaging studies call into question whether the results obtained using the b-CFS paradigm are genuinely attributable to high-level configural processing of the invisible face. Rather, they suggest that the representation of the perceptually suppressed face is limited to lower-level aspects such as its general shape. Therefore, in this study, we were interested to further study the representation of a perceptually suppressed face in a b-CFS context, capitalizing on the findings of a recent neuroimaging study. That is, *Chan et al. (2010)* recently showed that representations of body parts and faces were strongest in the extrastriate body area and fusiform face area, respectively, when they were presented in their commonly experienced configuration (e.g., the left side of a face presented in the right visual field). This result is intriguing since all conditions simply involved presenting the same stimulus (e.g., right or left side of a face) to a different side of the visual field. Thus, if stimulus strength influences suppression time, we would predict that perceptually suppressed face stimuli presented in their commonly experienced configuration would break suppression faster compared to those presented in the other part of the visual field. Moreover, given that the effect for the face stimuli seems to be specific to the fusiform face area, the presence of such an effect in a b-CFS setup could be indicative of the extent to which invisible face stimuli are processed during suppression. To this end, we also included a face inversion condition. That is, if a congruency effect is observed, this inversion condition will enable us to test whether this effect is dependent on specialized processing for upright faces.

## METHODS

### Participants

A total of 43 people participated in the experiment. All participants had normal or corrected-to-normal vision and were naïve with respect to the purposes of the study. The study was approved by the local ethics committee of the faculty (the Social and Societal Ethics Committee of the KU Leuven (SMEC) under the approval number G-2014 08 033). All participants provided written informed consent before the start of the experiment.

### Apparatus

Stimuli were shown on two 19.8-in. Sony Trinitron GDM F500-R (2048 × 1536 pixels at 60 Hz, for each) monitors driven by a DELL Precision T3400 computer with an Intel Core Quad CPU Q9300 2.5 GHz processor running on Windows XP. Binocular presentation was achieved by a custom made stereo set-up. Two CRT monitors, which stood opposite to each other (distance of 220 cm), projected to the left and right eye respectively via two mirrors placed at a distance of 110 cm from the screen. A head- and chin rest (15 cm from the mirrors) was used to stabilize fixation. The effective viewing distance was 125 cm. Stimulus presentation, timing and keyboard responses were

controlled with custom software programmed in Python using the PsychoPy library (*Peirce, 2007*; *Peirce, 2009*).

## Stimuli

The background of the display consisted of a random checkerboard pattern to achieve stable binocular fusion. The size of the individual elements of the checkerboard was equal to $0.34°$. In both eyes, a black frame ($10°$ by $10°$) was superimposed on the checkerboard pattern, onto which the stimuli would be presented. A black (eye dominance measurement) or white (main experiment) fixation cross was continuously present during the experiment (size 0.5 by $0.5°$). In the eye dominance measurement phase, the target consisted of an arrow (maximal width $4°$, maximal height $2°$) and the CFS mask consisted of 150 squares with randomly picked sizes between 1 and $2°$ and a random luminance value (range: 1–100 $cd/m^2$).

We obtained the stimuli used in the original study of *Chan et al. (2010)* and used a subset of those in this study (see Fig. 1A). That is, we only used the face configurations of their stimulus set, which consisted of four different half-face exemplars (size $3°$ of visual angle). For the specific details of the stimulus generation procedure, we refer to the original study. In the main experiment, the CFS mask ($6° \times 6°$) consisted of 200 grayscale squares with a random size between $0.75°$ and $1.5°$. In all parts of the experiments, the CFS mask refreshed its contents every 100 ms (i.e., at 10 Hz).

## Procedure

In the first part of the experiment, observers performed an eye dominance task according to the procedure outlined by *Yang, Blake and McDonald (2010)*. That is, on each trial, the CFS mask was presented to one of the observer's eyes and an arrow stimulus to the other eye. The arrow stimulus gradually increased from 0% to 100% contrast over a period of 2 seconds after which it remained present at full contrast. Upon breakthrough of the arrow stimulus, participants had to indicate as quickly as possible whether the arrow was pointing to the left or right. Participants performed this task for 80 trials in total (40 trials per eye). The dominant eye was determined by taking the eye for which the mean suppression was the lowest. In all subsequent phases of the experiment, the CFS mask was always presented to the dominant eye.

In the main part of the experiment each trial consisted of a 1 second fixation phase after which the CFS mask was presented to the dominant eye and the face stimulus to the non-dominant eye (Fig. 1B). The face stimulus gradually increased from 0% to 100% contrast in a period of 1 second after which it remained on screen at full contrast until the participants' response. Upon breakthrough, participants had to indicate as quickly as possible whether the face stimulus was presented to the left or right of fixation by means of a button press. Prior to the start of the main experiment, participants first completed a practice block to become acquainted with the task.

## Design

The experiment consisted of a $2 \times 2 \times 2$ full-factorial within-subjects design. Each stimulus (left or right side of a face) was presented in the left or right visual field in an

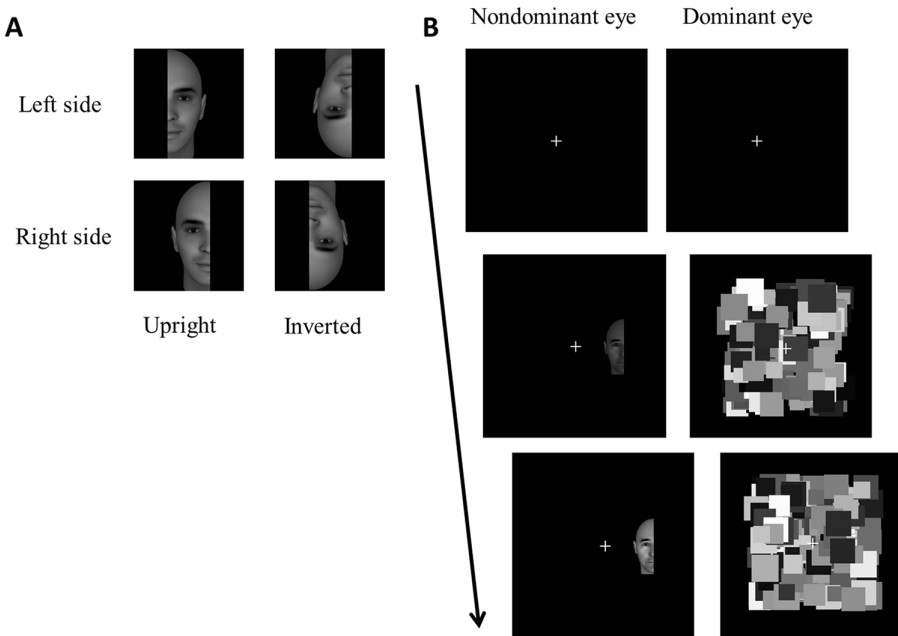

**Figure 1 Stimuli and procedure.** (A) Four different configurations for one face exemplar. Each configuration was presented either to the left or right side of the fixation cross. Presenting the top left stimulus to the right side of fixation would constitute an upright, congruent stimulus. (B) Trial sequence used in the experiment. Each trial started with a fixation period of 1 second after which the face stimulus was presented to the non-dominant eye and the CFS mask to the dominant eye. The face stimulus gradually increased in contrast and remained present at 100% contrast until the participants' response.

upright or inverted fashion. Participants completed a total of 96 trials. The practice block consisted of 8 trials.

## Data analysis

All analyses were performed in R, a statistical programming language (*R Core Team, 2014*). All statistical analyses were performed in a Bayesian framework, relying on model selection through Bayes Factors (*Rouder et al., 2009*; *Rouder et al., 2012*). In Bayesian statistics, statistical inference is performed by relying on Bayes' rule:

$$p(\theta|D) = \frac{p(D|\theta)p(\theta)}{p(D)}$$

where $\theta$ refers to a vector of parameters (e.g., the effect parameters of an ANOVA model) and $D$ to the data under consideration. In Bayes' rule, the prior probability distribution, $p(\theta)$, is then updated by the likelihood $p(D|\theta)$ to yield the posterior probability distribution, $p(\theta|D)$. In the Bayes Factor approach, the focus is on the marginal likelihood, $p(D)$:

$$p(D) = \int p(D|\theta)p(\theta)d\theta$$

The Bayes factor then refers to the ratio of marginal likelihoods of different statistical models under consideration (e.g., a model with main effects of congruency and inversion

versus a model with only a main effect of congruency), quantifying the change from prior to posterior model odds:

$$\frac{p(M_1|D)}{p(M_2|D)} = \frac{p(M_1)}{p(M_2)}\frac{p(D|M_1)}{p(D|M_2)}$$

where

$$BF_{12} = \frac{p(D|M_1)}{p(D|M_2)} = \frac{\int_{\theta_1} p(D|\theta)p(\theta)d\theta}{\int_{\theta_2} p(D|\theta)p(\theta)d\theta}$$

In itself, the Bayes Factor can be interpreted as a *relative* measure of evidence for one statistical model compared to another (e.g., a model with two main effects versus a model with two main effects and their interaction). That is, the value of the Bayes Factor has no absolute meaning, and should always be interpreted relative to the statistical models under consideration.

All Bayes Factors were computed using the R package BayesFactor version 0.9.11-1 (*Morey & Rouder, 2015*) using all default settings. The statistical models for which Bayes Factors were computed are akin to classical repeated measures ANOVA models, yet including random intercepts for both subject as well as stimulus (given that we used different face exemplars in our experiment, also known as a crossed random effects model: see *Clark, 1973*; *Baayen, Davidson & Bates, 2008*). Indeed, not taking into account the random stimulus effect can inflate the Type I error rate. *Rouder et al. (2012)* developed a default class of Bayes Factors for ANOVA designs and described the prior distributions used for calculating these Bayes Factors in detail. In short, normal distributions are used as priors for the fixed and random effects. These have a prior mean of zero, and an independent variance (width) for each of these effects, based on so-called g-priors developed by *Zellner and Siow (1980)*. The settings that can be adjusted in the BayesFactor package relate to the width of the prior distributions on the fixed and random effects (quantified by the scaling factor $r$). For the fixed effects we used the "wide" setting ($r = 0.5$) whereas for the random effects the "nuisance" option was used ($r = 1$). Following the classification proposed by *Jeffreys (1961)*, Bayes Factors >3 are considered to be convincing evidence for one model compared to another. In this paper, all Bayes Factors quantify how much more likely the best fitting model is compared to another model. That is, the best fitting model is always put in the numerator, whereas the other models under consideration are put in the denominator of the Bayes Factor equation.

## RESULTS

Before subjecting the data to any analysis, suppression times were first log transformed to account for their positive skew. Only correct responses were considered. Outliers were defined as suppression times that deviated more than three standard deviations from the mean suppression time (for each observer separately) and these were also excluded from the analysis. This led to a removal of 5.5% of the data. To facilitate the interpretation of the data, we converted the factors visual field and stimulus side to a single variable termed 'congruency.' A congruent stimulus would be one that constitutes a commonly

**Table 1 Bayes factor analysis.**

| Model | Bayes factor |
|---|---|
| Congruency + Inversion | 1 |
| Inversion | 3.6 |
| Congruency * Inversion | 5.2 |
| All other models | >100 |

**Notes:**
All Bayes Factors can be interpreted relative to the best fitting model (for which the Bayes Factor equals 1).
A + indicates that only main effects are included in the model.
A * denotes both main effects and the interaction between the conditions.

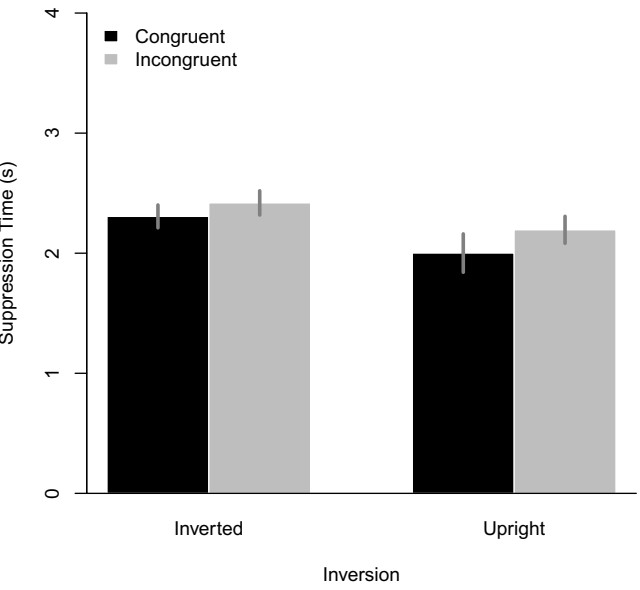

**Figure 2 Mean suppression times for all conditions.** Error bars denote 95% within-subject confidence intervals as described by *Morey (2008)*.

experienced configuration (e.g., right side of the face in the left visual field, assuming fixation in the center). For inverted stimuli, we applied the same transformation such that congruent stimuli would be the ones for which the overall configuration would be the same (e.g., an inverted left side of the face would now have to be presented in the left side of the visual field to be coded as congruent). Table 1 depicts the results of the Bayes Factor analysis. The betting fitting model (BF = 1) is one that includes a main effect of congruency and a main effect of inversion. This model is preferred 5.2 times over a model including also the interaction between the main effects. Furthermore, a model including only a main effect of inversion and no congruency effect is 3.6 times less likely than the best fitting model. For all other models (e.g., a model with a main effect of congruency only), the best fitting model was more than 100 more likely (i.e., BFs > 100). The mean suppression times for all combinations of congruency and face inversion are depicted in Fig. 2. In line with the Bayes Factor analysis, inverted faces yielded longer suppression times than upright faces (the well-known face inversion effect).

Furthermore, face stimuli presented in congruent configurations broke suppression faster than the incongruent ones, yet this main effect did was not modulated by stimulus inversion.

## DISCUSSION AND CONCLUSION

The goal of this study was to assess whether face stimuli presented in their commonly experienced configurations would break suppression faster than the same stimuli presented in other configurations. Our results indicated that this indeed was the case, yet the effect was not specific for upright face stimuli. That is, similar configurations also broke suppression faster when they were presented inverted rather than upright. This result implies that shape differences relative to fixation were responsible for the observed congruency effect rather than processing mechanisms specific for upright faces.

This study was motivated by the fact that a lot of b-CFS studies on face processing obtained evidence for relatively complex (high-level) processing of invisible faces during CFS. In contrast, studies relying on adaptation or neuroimaging techniques consistently showed that processing of invisible faces is severely reduced compared to visible faces and is possibly only specific to the general face shape rather than the identity, facial expression, or other high-level face attributes. Therefore, we decided to capitalize on the findings of a neuroimaging study in which it was shown that the pattern of responses in the fusiform face area was strongest for face stimuli presented in their commonly experienced (congruent) configuration. Assuming that stimuli with a strong representation break suppression faster, one would predict the same difference between congruent and incongruent configurations to be observed in a b-CFS setup. Moreover, given the specificity of the effect to the fusiform face area, we also predicted that the effect should be absent or at least greatly reduced for inverted faces (*Yovel & Kanwisher, 2005*). As highlighted above, our results indicated both an effect of configuration as well as inversion but no interaction between those factors. This indicates that the differences in suppression time between conditions are more likely attributable to shape-specific differences between conditions rather than mechanisms relying on the configural processing of faces, which are known to be affected by inversion (*Yin, 1969*; *Farah, Tanaka, & Drain, 1995*). Indeed, studies on holistic face perception have shown that face inversion is a stimulus manipulation that strongly influences performance on a wide range of tasks (for a review, see *Rossion, 2008*; *Van Belle et al., 2010*). Nevertheless, it has also been argued that inverted faces can still be processed holistically (*Richler et al., 2011*). This last study has mainly indicated qualitatively similar patterns for upright and inverted faces, but still observed quantitative differences. However, in our study the congruency effect was also quantitatively similar between upright and inverted faces, given the absence of an interaction between face inversion and congruency. Therefore, we think the most parsimonious explanation of our results is one that does not rely on face-specific (high-level) configural processing of perceptually suppressed face stimuli.

One particularly important difference between the stimuli presented in both types of configurations is the curvature of the face shape relative to fixation. That is, in congruent configurations, the curved contour is convex relative to fixation compared

to being concave in the incongruent configurations. Several behavioral studies have shown that convex features are often perceptually dominant, for instance, in determining figure-ground relationships or shape similarity (*Kanizsa & Gerbino, 1976*; *Bertamini & Wagemans, 2013*). Moreover, neurophysiological recordings have shown a similar bias towards convex features in macaque area V4 (*Pasupathy & Connor, 1999*). Last, a recent fMRI study has shown that cortical area LOC shows higher sensitivity for convex rather than concave shapes (*Haushofer et al., 2008*). Although our study only consisted of face stimuli, the pattern of results observed in this study is similar to what would be predicted based on a convexity/concavity account. Thus, in the light of these studies, we can speculate that our findings can be interpreted as potentially reflecting the heightened sensitivity of the visual system to convex features (relative to fixation).

This interpretation is in accord with a larger set of studies that has questioned evidence of high-level processing of stimuli suppressed through CFS. For example, *Hedger, Adams & Garner (2015a)* recently showed that the advantage of fearful faces breaking suppression faster than neutral ones is predicted by effective contrast of the stimuli. Furthermore, another recent study by the same group observed that attentional orienting due to threat stimuli is completely absent when threatening stimuli were rendered completely invisible (*Hedger, Adams & Garner, 2015b*). Other studies have cast doubt on whether invisible words can be processed (*Heyman & Moors, 2014*), numerosity can be extracted during suppression (*Liu et al., 2013*; *Hesselmann et al., 2014*; *Hesselmann & Knops, 2014*), or integration between a suppressed visual looming stimulus and a supraliminal auditory stimulus can occur (*Moors et al., 2015*).

In sum, the results of this study provide evidence that stimuli that are more strongly represented in the visual cortex break suppression faster than other stimuli. However, the fact that the observed congruency effect was not specific for upright face stimuli indicates that the face stimuli used in this study were presumably not processed by specialized face recognition mechanisms, but rather at a more basic level limited to more general shape properties such as convexity.

## ACKNOWLEDGEMENTS

We would like to thank David Boelens for assistance with data collection. We are grateful to Annie Chan for sharing the stimulus set and to the reviewers for their constructive suggestions.

### Funding

Pieter Moors and Lee de-Wit are respectively doctoral and postdoctoral fellows of the Research Foundation – Flanders (FWO). This work was supported in part by the Methusalem program by the Flemish Government (METH/14/02), awarded to Johan Wagemans. The funders had no role in study design, data collection and analysis, decision to publish, or preparation of the manuscript.

## Grant Disclosures

The following grant information was disclosed by the authors:

Research Foundation – Flanders: FWO.

Methusalem program by the Flemish Government: METH/14/02.

## Competing Interests

The authors declare that they have no competing interests.

## Author Contributions

- Pieter Moors conceived and designed the experiments, performed the experiments, analyzed the data, wrote the paper, prepared figures and/or tables, reviewed drafts of the paper.
- Johan Wagemans wrote the paper, reviewed drafts of the paper.
- Lee de-Wit conceived and designed the experiments, wrote the paper, reviewed drafts of the paper.

## Human Ethics

The following information was supplied relating to ethical approvals (i.e., approving body and any reference numbers):

Social and Societal Ethics Committee of the KU Leuven (SMEC) approval: G-2014 08 033.

## Data Deposition

FigShare: https://dx.doi.org/10.6084/m9.figshare.2055867.v1

## Supplemental Information

Supplemental information for this article can be found online at http://dx.doi.org/10.7717/peerj.1565#supplemental-information.

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
