# Peer review of "Faces in commonly experienced configurations enter awareness faster due to their curvature relative to fixation"

_PeerJ, doi:10.7717/peerj.1565_

## Round 0.1 · original submission · Major Revisions

The reviewers are largely united in their major suggestions. I think this will make for a nice, short, and clear study once revised to the reviewers' satisfaction.

Reviewer 1 ·

Basic reporting

The main comment I have is that the methods (and the interpretation of the results) describing the Bayes Factor analyses are practically non-existent. Indeed, there is only a brief description about them in the Results section. The paper relies very heavily on these analyses (i.e. there is only one table, one data figure, and no other statistics) but gives very little description about the details behind the method. I think some description about the chosen prior probability distributions, the theory behind Bayes factor analyses, perhaps a sample calculation showing how the factors are computed for their data, and more detailed description of how one should interpret the Bayes Factors in the context of this study. Also, the relationship between Figure 2 and Table 1 should be made clearer.

Experimental design

No Comments

Validity of the findings

No Comments

Additional comments

MAJOR COMMENTS

Comment 1
The main comment I have is that the methods (and the interpretation of the results) describing the Bayes Factor analyses are practically non-existent. Indeed, there is only a brief description about them in the Results section. The paper relies very heavily on these analyses (i.e. there is only one table, one data figure, and no other statistics) but gives very little description about the details behind the method. I think some description about the chosen prior probability distributions, the theory behind Bayes factor analyses, perhaps a sample calculation showing how the factors are computed for their data, and more detailed description of how one should interpret the Bayes Factors in the context of this study. Also, the relationship between Figure 2 and Table 1 should be made clearer.

Comment 2
On line 221, the authors say: “In the light of these studies, our findings can be interpreted as reflecting the heightened sensitivity of the visual system to convex features (relative to fixation).” This is a strong statement that they have provided little evidence for. It should be toned down.

MINOR COMMENTS

Comment 1
Line 39: “for a time period in the order of seconds” should be “for a time period ON the order of seconds”

Comment 2
Line 120: “with a randomly picked sizes” should be “with a randomly picked size”

Comment 3
Line 120: The authors say “random luminance value.”… What is the range?

Comment 4
In table 1, I believe “> 100” should be “< 100” (i.e. Bayes Factor less than 100). Please double check.

Comment 5
Line 196: The authors say “Our results indicated that this indeed was the case, yet that the effect was not specific for” but it should be “Our results indicated that this indeed was the case, yet the effect was not specific for”

Comment 6
Table 1 should come after Figure 2 (as it is cited after Figure 2)

Reviewer 2 ·

Basic reporting

No comments

Experimental design

No comments

Validity of the findings

No comments

Additional comments

The paper of Moors et al., explored whether invisible faces presented in the visual field which we are used to see them are processed differently compared to when they presented in other visual field. They indeed found a difference and interestingly, the result did not depend on whether face was upright or inverted. The authors interpret the effect as related to shape processing (concave vs. convex). I think it is a nice and simple study. The results are in general sound. Having said that, I would like to make several suggestions to improve the manuscript.

Major

1. Introduction: The interpretation of the breaking CFS as the authors present (line 43-47) is not that straightforward. For example, (Mudrik et al., 2011) showed that " Complex scenes that included incongruent objects escaped perceptual suppression faster than normal scenes did.". I do not think we really know why it breaks to suppression faster. So, I suggest that you write more accurately.

2. Statistic analysis
a. As far as I understand, for calculating Bayes factor one needs to define a prior. It was not clear from the paper what prior was selected.
b. I am not sure I understood what exactly were three models (Table 1). In particular, what is the difference between '+' and '*'. If we parallel the Bayesian statistics to "classical", I would expect to see two main effects and interaction between them.
c. What is the interpretation of significant (>3) inversion effect?
d. It would be interesting to see the results of "classical" repeated-measures ANOVA
e. Was the any effect of visual field of the presentation? The authors show only collapsed the data across both visual fields.
f. Was the difference between inverted and upright face effects significant (collapsed across congruent & incongruent) ?

3. The fact that using inverted & upright faces the authors obtained similar congruency effect indeed favor the interpretation that that the effect was modulated by non-face specific mechanisms. Ideally, to support convex/concave hypothesis, the authors had to run control experiment with such non-face stimuli. Given that they did do that, theoretically this is possible that inverted face was still processed as a face. First, during 2 sec till awareness break the subjects could make a mental rotation. Second, some claim that inverted faces might still be processed similarly to upright faces (Richler et al., 2011). In any case, I suggest that the authors discuss the literature about inverted faces and why they are likely not to be processed differently from upright faces.

Minor

1. Literature suggestions:

a. Line 49: I think a recent review of (Axelrod et al., 2015) is very relevant to what you explore.
b. Line 59: You can add also trustworthiness which was tested (Stewart et al., 2012)
c. Line 60: There was perceptual groping papers not in the context of face (Wang et al., 2012) and again (Mudrik et al., 2011). I think that non-face perceptual groping is worth to be discussed or at least mentioned.
d. Line 212: Regarding low-level processing and CFS, there was also relevant paper of Yang and Blake (Yang and Blake, 2012)

2. There is no data analysis section in Methods. I think that most of information about Bayesian analysis is better to be moved there.

3. Hand dominance of the participants was not specified

4. Line 124: please specify whether you received the stimuli from Chan et al or generated them from scratch. If the latter, I would suggest that you still provide some details regarding stimuli generation.

5. Line 180: “slower suppression time” is not a clear term. I think longer / shorter would be a more clear terminology.

6. Line 197: either add ‘also’ word or describe also upright conditions. Currently, this sentence is misleading because it appears as though the effect was found only for inverted faces.


References

Axelrod V, Bar M, Rees G (2015) Exploring the unconscious using faces. Trends in Cognitive Sciences 19:35-45.
Mudrik L, Breska A, Lamy D, Deouell LY (2011) Integration Without Awareness Expanding the Limits of Unconscious Processing. Psychol Sci 22:764-770.
Richler JJ, Mack ML, Palmeri TJ, Gauthier I (2011) Inverted faces are (eventually) processed holistically. Vision Res 51:333-342.
Stewart LH, Ajina S, Getov S, Bahrami B, Todorov A, Rees G (2012) Unconscious evaluation of faces on social dimensions. J Exp Psychol Gen 141:715.
Wang L, Weng X, He S (2012) Perceptual grouping without awareness: Superiority of kanizsa triangle in breaking interocular suppression. PLoS One 7:e40106.
Yang E, Blake R (2012) Deconstructing continuous flash suppression. J Vis 12.

---

## Round 0.2 · accepted · Accept

The authors responded to the reviewers' concerns satisfactorily. Please address the remaining minor concern in the production phase.

Reviewer 1 ·

Basic reporting

No Comments

Experimental design

No Comments

Validity of the findings

No Comments

Additional comments

No Comments

Reviewer 2 ·

Basic reporting

No comments

Experimental design

No comments

Validity of the findings

No comments

Additional comments

I thank the authors for addressing my concerns of Rev1.

My only suggestion now is that the authors include their motivation of not using classical ANOVA within the paper (e.g., Methods section). In particular, if they think that classical ANOVA is technically incorrect in their case (as they wrote it now in the rebuttal), then I suggest that they explain this in the paper (preferably with more details than it is now). The choice of statistical method is an important part of the research, therefore it is essential to provide such motivation.